# Confusions over Time: An Interpretable Bayesian Model to Characterize Trends in Decision Making

**Himabindu Lakkaraju**
Department of Computer Science
Stanford University
himalv@cs.stanford.edu

**Jure Leskovec**
Department of Computer Science
Stanford University
jure@cs.stanford.edu

## Abstract

We propose *Confusions over Time* (CoT), a novel generative framework which facilitates a multi-granular analysis of the decision making process. The CoT not only models the confusions or error properties of individual decision makers and their evolution over time, but also allows us to obtain diagnostic insights into the collective decision making process in an interpretable manner. To this end, the CoT models the confusions of the decision makers and their evolution over time via time-dependent confusion matrices. Interpretable insights are obtained by grouping *similar* decision makers (and items being judged) into clusters and representing each such cluster with an appropriate *prototype* and identifying the most important features characterizing the cluster via a *subspace feature indicator* vector. Experimentation with real world data on bail decisions, asthma treatments, and insurance policy approval decisions demonstrates that CoT can accurately model and explain the confusions of decision makers and their evolution over time.

## 1 Introduction

Several diverse domains such as judiciary, health care, and insurance rely heavily on human decision making. Since decisions of judges, doctors, and other decision makers impact millions of people, it is important to reduce errors in judgement. The first step towards reducing such errors and improving the quality of decision making is to diagnose the errors being made by the decision makers. It is crucial to not only identify the errors made by individual decision makers and how they change over time, but also to determine common patterns of errors encountered in the collective decision making process. This turns out to be quite challenging in practice partly because there is no ground truth which captures the optimal decision in a given scenario.

Prior research has mainly focussed on modeling decisions of individual decision makers [2, 12]. For instance, the Dawid-Skene model [2] assumes that each *item* (eg., a patient) has an underlying *true label* (eg., a particular treatment) and a decision maker's evaluation of the item will be masked by her own biases and confusions. Confusions of each individual decision maker $j$ are modeled using a latent confusion matrix $\Theta_j$, where an entry in the $(p, q)$ cell denotes the probability that an item with true label $p$ will be assigned a label $q$ by the decision maker $j$. The true labels of items and latent confusion matrices of decision makers are jointly inferred as part of the inference process. However, a major drawback of the Dawid-Skene framework and several of its extensions [16, 17, 12] is that they do not provide any diagnostic insights into the collective decision making process. Furthermore, none of these approaches account for temporal changes in the confusions of decision makers.

Here, we propose a novel Bayesian framework, Confusions over Time (CoT), which jointly: 1) models the *confusions* of individual decision makers 2) captures the *temporal dynamics* of their decision making 3) provides *interpretable insights* into the collective decision making process, and 4)

infers *true labels* of items. While there has been prior research on each of the aforementioned aspects independently, there has not been a single framework which ties all of them together in a principled yet simple way.

The modeling process of the CoT groups decision makers (and items) into clusters. Each such cluster is associated with a *subspace feature indicator* vector which determines the most important features that characterize the cluster, and a *prototype* which is the representative data point for that cluster. The prototypes and the subspace feature indicator vectors together contribute to obtaining interpretable insights into the decision making process. The decisions made by decision makers on items are modeled as interactions between such clusters. More specifically, each pair of (decision maker, item) clusters is associated with a set of latent confusion matrices, one for each discrete time instant. The decisions are then modeled as multinomial variables sampled from such confusion matrices. The inference process involves jointly inferring cluster assignments, the latent confusion matrices, prototypes and feature indicator vectors corresponding to each of the clusters, and true labels of items using a collapsed Gibbs sampling procedure.

We analyze the performance of CoT on three real-world datasets: (1) judicial bail decisions; (2) treatment recommendations for asthma patients; (3) decisions to approve/deny insurance requests. Experimental results demonstrate that the proposed framework is very effective at inferring true labels of items, predicting decisions made by decision makers, and providing diagnostic insights into the collective decision making process.

## 2   Related Work

Here, we provide an overview of related research on modeling decision making. We also highlight the connections of this work to two other related yet different research directions: stochastic block models, and interpretable models.

**Modeling decision making.** There has been a renewed interest in analyzing and understanding human decisions due to the recent surge in applications in crowdsourcing, public policy, and education [6]. Prior research in this area has primarily focussed on the following problems: inferring true labels of items from human annotations [17, 6, 18, 3], inferring the expertise of decision makers [5, 19, 21], analyzing confusions or error properties of individual decision makers [2, 12, 10], and obtaining diagnostic insights into the collective decision making process [10].

While some of the prior work has addressed each of these problems independently, there have been very few attempts to unify the aforementioned directions. For instance, Whitehill et. al. [19] proposed a model which jointly infers the true labels and estimate of evaluator's quality by modeling decisions as functions of the expertise levels of decision makers and the difficulty levels of items. However, this approach neither models the error properties of decision makers, nor provides any diagnostic insights into the process of decision making. Approaches proposed by Skene *et al.* [2] and Liu *et al.* [12] model the confusions of individual decision makers and also estimate the true labels of items, but fail to provide any diagnostic insights into the patterns of collective decisions. Recently, Lakkaraju *et al.* proposed a framework [10] which also provides diagnostic insights but it requires a post-processing step employing Apriori algorithm to obtain these insights. Furthermore, none of the aforementioned approaches model the temporal dynamics of decision making.

**Stochastic block models.** There has been a long line of research on modeling relational data using stochastic block models [15, 7, 20]. These modeling techniques typically involved grouping entities (eg., nodes in a graph) such that interactions between these entities (eg., edges in a graph) are governed by their corresponding clusters. However, these approaches do not model the nuances of decision making such as confusions of decision makers which is crucial to our work.

**Interpretable models.** A large body of machine learning literature focused on developing interpretable models for classification [11, 9, 13, 1] and clustering [8]. To this end, various classes of models such as decision lists [11], decision sets [9], prototype (case) based models [1], and generalized additive models [13] were proposed. However, none of these approaches can be readily applied to determine error properties of decision makers.

# 3  Confusions over Time Model

In this section, we present CoT, a novel Bayesian framework which facilitates an interpretable, multi-granular analysis of the decision making process. We begin by discussing the problem setting and then dive into the details of modeling and inference.

## 3.1  Setting

Let $J$ and $I$ denote the sets of decision makers and items respectively. Each item is judged by one or more decision makers. In domains such as judiciary and health care, each defendant or patient is typically assessed by a single decision maker, where as each item is evaluated by multiple decision makers in settings such as crowdsourcing. Our framework can handle either scenarios and does not make any assumptions about the number of decision makers required to evaluate an item. However, we do assume that each item is judged no more than once by any given decision maker. The decision made by a decision maker $j$ about an item $i$ is denoted by $r_{i,j}$. Each decision $r_{i,j}$ is associated with a discrete time stamp $t_{i,j} \in \{1, 2 \cdots T\}$ corresponding to the time instant when the item $i$ was evaluated by the decision maker $j$.

Each decision maker has $M$ different features or attributes and $a_m^{(j)}$ denotes the value of the $m^{th}$ feature of decision maker $j$. Similarly, each item has $N$ different features or attributes and $b_n^{(i)}$ represents the value of the $n^{th}$ feature of item $i$. Each item $i$ is associated with a true label $z_i \in \{1, 2, \cdots K\}$. $z_i$ is not observed in the data and is modeled as a latent variable. This mimics most real-world scenarios where the ground-truth capturing the optimal decision or the true label is often not available.

## 3.2  Defining Confusions over Time (CoT) model

The CoT model jointly addresses the problems of modeling confusions of individual decision makers and how these confusions change over time, and also provides interpretable diagnostic insights into the collective decision making process. Each of these aspects is captured by the generative process of the CoT model, which comprises of the following components: (1) cluster assignments; (2) prototype selection and subspace feature indicator generation for each of the clusters; (3) true label generation for each of the items; (4) time dependent confusion matrices. Below, we describe each of these components and highlight the connections between them.

***Cluster Assignments.***    The CoT model groups decision makers and items into clusters. The model assumes that there are $L_1$ decision maker clusters and $L_2$ item clusters. The values of $L_1, L_2$ are assumed to be available in advance. Each decision maker $j$ is assigned to a cluster $c_j$, which is sampled from a multinomial distribution with a uniform Dirichlet prior $\epsilon_\alpha$. Similarly, each item $i$ is associated with a cluster $d_i$, sampled from a multinomial distribution with a uniform Dirichlet prior $\epsilon_{\alpha'}$. The features of decision makers and the decisions that they make depend on the clusters they belong to. Analogously, the true labels of items, their features and the decisions involving them are influenced by the clusters to which they are affiliated.

***Prototype and Subspace Feature Indicator.***    The interpretability of the CoT stems from the following two crucial components: associating each decision maker and item cluster with a *prototype* or an exemplar, and a subspace feature indicator which is a binary vector indicating which features are important in characterizing the cluster. The prototype $p_c$ of a decision maker cluster $c$ is obtained by sampling uniformly over all decision makers $1 \cdots |J|$ i.e., $p_c \sim Uniform(1, |J|)$. The subspace feature indicator, $\omega_c$, of the cluster $c$ is a binary vector of length $M$. An element of this vector, $\omega_{c,f}$, corresponds to the feature $f$ and indicates if that feature is important ($\omega_{c,f} = 1$) in characterizing the cluster $c$. $\omega_{c,f} \in \{0, 1\}$ is sampled from a Bernoulli distribution. The prototype $p_d'$ and subspace feature indicator vector $\omega_d'$ corresponding to an item cluster $d$ are defined analogously.

*Generating the features:* The prototype and the subspace feature indicator vector together provide a template for generating feature values of the cluster members. More specifically, if the $m^{th}$ feature is designated as an important feature for cluster $c$, then instances in that cluster are very likely to inherit the coresponding feature value from the prototype datapoint $p_c$.

We sample the value of a discrete feature $m$ corresponding to decision maker $j$, $a_m^{(j)}$, from a multinomial distribution $\phi_{c_j,m}$ where $c_j$ denotes the cluster to which $j$ belongs. $\phi_{c_j,m}$ is in turn sampled from a Dirichlet distribution parameterized by the vector $g_{p_{c_j,m},\omega_{c_j,m},\lambda}$ i.e., $\phi_{c_j,m} \sim Dirichlet(g_{p_{c_j,m},\omega_{c_j,m},\lambda})$. $g_{p_{c,m},\omega_{c,m},\lambda}$ is a vector defined such that the $e^{th}$ element of the vector corresponds to the prior on the $e^{th}$ possible value of the $m^{th}$ feature. The $e^{th}$ element of this vector is defined as:

$$g_{p_{c,m},\omega_{c,m},\lambda}(e) = \lambda \left(1 + \mu \mathbb{1}\left[\omega_{c,m} = 1 \text{ and } p_{c,m} = V_{m,e}\right]\right) \tag{1}$$

where $\mathbb{1}$ denotes the indicator function, and $\lambda$ and $\mu$ are the hyperparameters which determine the extent to which the prototype will be copied by the cluster members. $V_{m,e}$ denotes the $e^{th}$ possible value of the $m^{th}$ feature. For example, let us assume that the $m^{th}$ feature corresponds to gender which can take one of the following values: $\{male, female, NA\}$, then $V_{m,1}$ represents the value $male$, $V_{m,2}$ denotes the values $female$ and so on.

Equation 1 can be explained as follows: if the $m^{th}$ feature is irrelevant to the cluster $c$ (i.e., $\omega_{c,m} = 0$), then $\phi_{c,m}$ will be sampled from a Dirichlet distribution with a uniform prior $\lambda$. On the other hand, if $\omega_{c,m} = 1$, then $\phi_{c,m}$ has a prior of $\lambda + \mu$ on that feature value which matches the prototype's feature value, and a prior of just $\lambda$ on all the other possible feature values. The larger the value of $\mu$, the higher the likelihood that the cluster members assume the same feature value as that of the prototype.

Values of continuous features are sampled in an analogous fashion. We model continuous features as Gaussian distributions. If a particular continuous feature is designated as an important feature for some cluster $c$, then the mean of the Gaussian distribution corresponding to this feature is set to be equal to that of the corresponding feature value of the prototype $p_c$, otherwise the mean is set to 0. The variance of the Gaussian distribution is set to $\sigma$ for all continuous features.

Though the above exposition focused on clusters of decision makers, we can generate feature values of items in a similar manner. Feature values of items belonging to some cluster $d$ are sampled from the corresponding feature distributions $\phi'_d$, which are in turn sampled from priors which account for the prototype $p'_d$ and subspace feature indicator $\omega'_d$.

***True Labels of Items.*** Our model assumes that every item $i$ is associated with a true label $z_i$. This true label is sampled from a multinomial distribution $\rho_{d_i}$ where $d_i$ is the cluster to which $i$ belongs. $\rho_{d_i}$ is sampled from a Dirichlet prior which ensures that the true labels of the members of cluster $d_i$ conform to the true label of the prototype. The prior is defined using a vector $g'_{p'_d}$ and each element of this vector can be computed as:

$$g'_{p'_d}(e) = \lambda \left(1 + \mu \mathbb{1}\left[z_{p'_d} = e\right]\right) \tag{2}$$

Note that Equation 2 assigns a higher prior to the label which is the same as that of the cluster's prototype. The larger the value of $\mu$, the higher the likelihood that the true labels of all the cluster members will be the same as that of the prototype.

***Time Dependent Confusion Matrices.*** Each pair of decision maker-item clusters $(c, d)$ is associated with a set of latent confusion matrices $\Theta_{c,d}^{(t)}$, one for each discrete time instant $t$. These confusion matrices influence how the decision makers in the cluster $c$ judge the items in $d$ and also allow us to study how decision maker confusions change with time.

Each confusion matrix is of size $K \times K$ where $K$ denotes the number of possible values that an item can be labeled with. Each entry $(p, q)$ in a confusion matrix $\Theta$ determines the probability that an item with true label $p$ will be assigned the label $q$. Higher probability mass on the diagonal signifies accurate decisions.

Let us consider the confusion matrix corresponding to decision maker cluster $c$, item cluster $d$, and time instant 1 (first time instant): $\Theta_{c,d}^{(1)}$. Each row of this matrix denoted by $\Theta_{c,d,z}^{(1)}$ ($z$ is the row index) is sampled from a Dirichlet distribution with a uniform prior $\wedge$. The $CoT$ framework also models the dependencies between the confusion matrices at consecutive time instants via a trade-off parameter $\pi$. The magnitude of $\pi$ determines how similar $\Theta_{c,d}^{(t+1)}$ is to $\Theta_{c,d}^{(t)}$. The $e^{th}$ element in the row $z$ of the confusion matrix $\Theta_{c,d,z}^{(t+1)}$ is sampled as follows:

$$\Theta_{c,d,z}^{(t+1)}(e) \sim Dirichlet(h_{\Theta_{c,d,z}^{(t)}(e),\wedge}) \text{ where } h_{\Theta_{c,d,z}^{(t)}(e),\wedge} = \wedge \left(1 + \pi \left[\Theta_{c,d,z}^{(t)}(e)\right]\right) \tag{3}$$

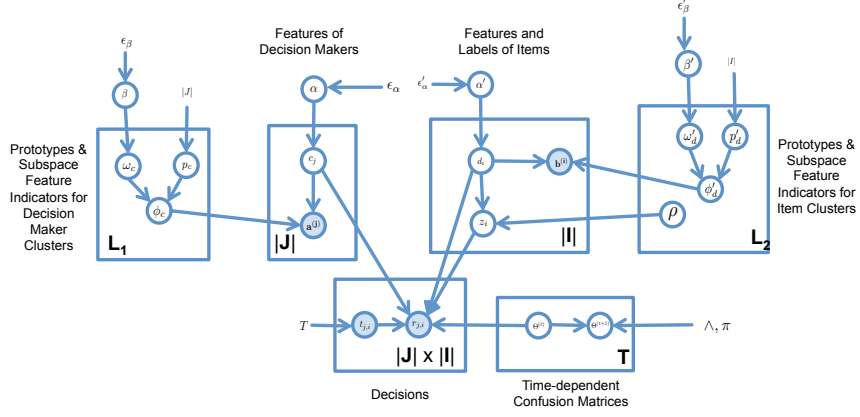

Figure 1: Plate notation for the CoT model. Each block is annotated with descriptive text. The hyperparameters $\lambda, \mu$ are omitted to improve readability.

*Generating the decisions:* Our model assumes that the decision $r_{j,i}$ made by a decision maker $j$ about an item $i$ depends on the clusters $c_j, d_i$ that $j$ and $i$ belong to, the time instant $t_{j,i}$ when the decision was made, and the true label $z_i$ of item $i$. More specifically, $r_{j,i} \sim Multinomial(\Theta_{c_j,d_i,z_i}^{(t_{i,j})})$.

***Complete Generative Process.*** Please refer to the Appendix for the complete generative process of CoT. The graphical representation of CoT is shown in Figure 1.

### 3.3 Inference

We use collapsed Gibbs sampling [4] approach to infer the latent variables of the CoT framework. This technique involves integrating out all the intermediate latent variables $\Theta, \phi, \phi', \rho$ and sampling only the variables coresponding to prototypes $p_c, p'_d$, subspace feature indicator vectors $\omega_{c,m}, \omega'_{d,n}$, cluster assignments $c_j, d_i$ and item labels $z_i$. The update equation for $p_c$ is given by: [1].

$$p(p_c = q|\mathbf{z}, \mathbf{c}, \mathbf{d}, \boldsymbol{\omega}, \boldsymbol{\omega}', \mathbf{p}', \mathbf{p}^{(-c)}) \propto \prod_{m=1}^{M} \mathbb{1}\left(\omega_{c,m} = 1\right) \times u_{c,m,q_m} \qquad (4)$$

where $u_{c,m,q_m}$ denotes the number of instances belonging to cluster $c$ for which the discrete feature $m$ takes the value $q_m$. $q_m$ denotes the value of the feature $m$ corresponding to the decision maker $q$. The update equation for $p'_d$ can be derived analogously. The conditional distribution for $\omega_{c,m}$ obtained by integrating out $\phi$ variables is:

$$p(\omega_{c,m} = s|\mathbf{z}, \mathbf{c}, \mathbf{d}, \boldsymbol{\omega}^{-(c,m)}, \boldsymbol{\omega}', \mathbf{p}', \mathbf{p}, \lambda) \propto \begin{cases} \beta \times \frac{\boldsymbol{B}(g_{p_c,m,1,\lambda+\tilde{u}_c})}{\boldsymbol{B}(g_{p_c,m,1,\lambda})} & \text{if } s = 1 \\ (1-\beta) \times \frac{\boldsymbol{B}(g_{p_c,m,0,\lambda+\tilde{u}_c})}{\boldsymbol{B}(g_{p_c,m,0,\lambda})} & \text{otherwise} \end{cases} \qquad (5)$$

where $\tilde{u}_c$ denotes the number of decision makers belonging to cluster $c$ and $\boldsymbol{B}$ denotes the Beta function. The conditional distribution for $\omega'_{d,n}$ can be written in an analogous manner. The conditional distributions for $c_j, d_i$ and $z_i$ can be derived as described in [10].

## 4 Experimental Evaluation

In this section, we present the evaluation of the CoT model on a variety of datasets. Our experiments are designed to evaluate the performance of our model on a variety of tasks such as recovering confusion matrices, predicting item labels and decisions made by decision makers. We also study the interpretability aspects of our model by evaluating the insights obtained.

| Dataset | # of Evaluators | # of Items | # of Decisions | Evaluator Features | Item Features |
|---------|-----------------|------------|----------------|--------------------|---------------|
| Bail | 252 | 250,500 | 250,500 | # of felony, misd., minor offense cases | Previous arrests, offenses, pays rent, children, gender |
| Asthma | 48 | 60,048 | 62,497 | Gender, age, experience, specialty, # of patients seen | Gender, age, asthma history, BMI, allergies |
| Insurance | 74 | 49,876 | 50,943 | # of policy decisions, # of construction, chemical, technology decisions | domain, previous losses, premium amount quoted |

Table 1: Summary statistics of our datasets.

**Datasets.** We evaluate CoT on the following real-world datasets: (1) **Bail dataset** comprising of information about criminal court judges deciding if defendants should be released without any conditions, asked to pay bail amount, or be locked up ($K = 3$); Here, decision makers are judges and items are defendants. (2) **Asthma dataset** which captures the treatment recommendations given by doctors to patients. Patients are recommended one of the two possible categories of treatments: mild (mild drugs/inhalers), strong (nebulizers/immunomodulators) ($K = 2$). (3) **Insurance data** which contains information about client managers deciding if a client company's insurance request should be approved or denied ($K = 2$). Each of the datasets spans about three years in time. We do have the ground-truth of true labels associated with defendants/patients/insurance clients in the form of expert decisions and observed consequences for each of the datasets. Note, however, that we only use a small fraction (5%) of the available true labels during the learning process. The decision makers and items are associated with a variety of features in each of these datasets (Table 1).

**Baselines & Ablations.** We benchmark the performance of CoT against the following state-of-the-art baselines: Dawid-Skene Model (DS) [2], Single Confusion model (SC) [12], Hybrid Confusion Model (HC) [12], Joint Confusion Model (JC) [10]. DS, SC and HC models focus only on modeling decisions of individual decision makers and do not provide any diagnostic insights into the decision making process JC model, on the other hand, also provides diagnostic insights (via post processing) None of the baselines account for the temporal aspects.

To evaluate the importance of the various components of our model, we also consider multiple ablations of CoT. *Non-temporal CoT (NT-CoT)* is a variant of CoT which does not incorporate the temporal component and hence is applicable to a single time instance. *Non-intepretable CoT (NI-CoT)* is another ablation which does not involve the prototype or subspace feature indicator vector generation, instead $\phi$, $\phi'$, and $\rho$ are sampled from symmetric Dirichlet priors.

**Experimental Setup.** In most real-world settings involving human decision making, the true labels of items are available for very few instances. We mimic this setting in our experiments by employing *weak supervision*. We let the all models (including the baselines) access the true labels of about 5% of the items (selected randomly) in the data during the learning phase. In all of our experiments, we divide each dataset into three discrete time chunks. Each time chunk corresponds to a year in the data. While our model can handle the temporal aspects explicitly, the same is not true for any of the baselines as well as the ablation Non-temporal CoT. To work around this, we run each of these models separately on data from each time slice. We run the collapsed Gibbs sampling inference until the approximate convergence of log-likelihood. All the hyperparameters of our model are initialized to standard values: $\epsilon_\beta = \epsilon'_\beta = \wedge = \pi = \epsilon_\alpha = \epsilon'_\alpha = \lambda = \mu = 1, \sigma = 0.1$. The number of decision maker and item clusters $L_1$ and $L_2$ were set using the Bayesian Information Criterion (BIC) metric [14]. The parameters of all the other baselines were chosen similarly.

## 4.1 Evaluating Estimated Confusion Matrices and Predictive Power

We evaluate CoT on estimating confusion matrices, predicting item labels, and predicting decisions of decision makers. We first present the details of each task and then discuss the results.

**Recovering Confusion Matrices.** We experiment with the CoT model to determine how accurately it can recover decision maker confusion matrices. To measure this, we use the Mean Absolute Error

| Task | Predicting item labels | | | Inferring confusion matrices | | | Predicting decisions | | |
|---|---|---|---|---|---|---|---|---|---|
| Method | Bail | Asthma | Insurance | Bail | Asthma | Insurance | Bail | Asthma | Insurance |
| SC | 0.53 | 0.59 | 0.51 | 0.38 | 0.31 | 0.40 | 0.52 | 0.51 | 0.55 |
| DS | 0.61 | 0.63 | 0.64 | 0.32 | 0.28 | 0.36 | 0.56 | 0.58 | 0.58 |
| HC | 0.62 | 0.65 | 0.66 | 0.31 | 0.26 | 0.33 | 0.59 | 0.64 | 0.61 |
| JC | 0.64 | 0.68 | 0.69 | 0.26 | 0.19 | 0.29 | 0.64 | 0.67 | 0.66 |
| LR | 0.56 | 0.60 | 0.57 | | | | 0.58 | 0.60 | 0.57 |
| NT-CoT | 0.65 | 0.68 | 0.69 | 0.24 | 0.19 | 0.28 | 0.66 | 0.68 | 0.66 |
| NI-CoT | 0.69 | 0.70 | 0.70 | 0.21 | 0.18 | 0.26 | 0.67 | 0.70 | 0.68 |
| **CoT** | 0.71 | 0.72 | 0.74 | 0.19 | 0.16 | 0.23 | 0.69 | 0.74 | 0.71 |
| Gain % | 9.86 | 5.56 | 6.76 | 36.84 | 18.75 | 26.09 | 7.25 | 9.46 | 7.04 |

Table 2: Experimental results: CoT consistently performs best across all tasks and datasets. Bottom row of the table indicates percentage gain of the CoT over the best performing baseline JC.

(MAE) metric to compare the elements of the estimated confusion matrix ($\Theta'$) and the observed confusion matrix ($\Theta$). MAE of two such matrices is the sum of element wise differences:

$$MAE(\Theta, \Theta') = \frac{1}{K^2} \sum_{u=1}^{K} \sum_{v=1}^{K} |\Theta_{u,v} - \Theta'_{u,v}|$$

While the baseline models SC, DS, HC associate a single confusion matrix with each decision maker, the baseline JC and our model assume that each decision maker can have multiple confusion matrices (one per each item cluster). To ensure a fair comparison, we apply the MAE metric every time a decision maker judges an item choosing the appropriate confusion matrix and then compute the average MAE.

**Predicting Item Labels.** We also evaluate the CoT on the task of predicting item labels. We use the AUC ROC metric to measure the predictive performance. In addition to the previously discussed baselines, we also compare the performance against Logistic Regression (LR) classifier. The LR model was provided decision maker and item features, time stamps, and decision maker decisions as input features.

**Predicting Evaluator Decisions.** The CoT model can also be used to predict decision maker decisions. Recall that the decision maker decisions are regarded as observed variables through out the inference. However, we can leverage the values of all the latent variables learned during inference to predict the decision maker decisions. In order to execute this task, we divide the data into 10 folds and carry out the Gibbs sampling inference procedure on the first 9 folds where the decision maker decisions are observed. We then use the estimated latent variables to sample the decision maker decisions for the remaining fold. We repeat this process over each of the 10 folds and report average AUC.

**Results and Discussion.** Results of all the tasks are presented in Table 2. CoT outperforms all the other baselines and ablations on all the tasks. The SC model which assumes that all the decision makers share a single confusion matrix performs extremely poorly compared to the other baselines, indicating that its assumptions are not valid in real-world datasets. The JC model, which groups *similar* decision makers and items together turns out to be one of the best performing baseline. The performance of our ablation models indicates that excluding the temporal aspects of the CoT causes a dip in the performance of the model on all the tasks. Furthermore, leaving out the interpretability components affects the model performance slightly. These results demonstrate the utility of the joint inference of temporal, interpretable aspects alongside decision maker confusions and cluster assignments.

## 4.2 Evaluating Interpretability

In this section, we first present an evaluation of the quality of the clusters generated by the CoT model. We then discuss some of the qualitative insights obtained using CoT.

| Model | Purity | | | Inverse Purity | | |
|---|---|---|---|---|---|---|
| | Bail | Asthma | Insurance | Bail | Asthma | Insurance |
| JC | 0.67 | 0.71 | 0.74 | 0.63 | 0.66 | 0.67 |
| NT-CoT | 0.74 | 0.78 | 0.79 | 0.72 | 0.73 | 0.76 |
| NI-CoT | 0.72 | 0.76 | 0.78 | 0.67 | 0.71 | 0.72 |
| **CoT** | 0.83 | 0.84 | 0.81 | 0.78 | 0.79 | 0.82 |

Table 3: Average purity and inverse purity computed across all decision maker and item clusters.

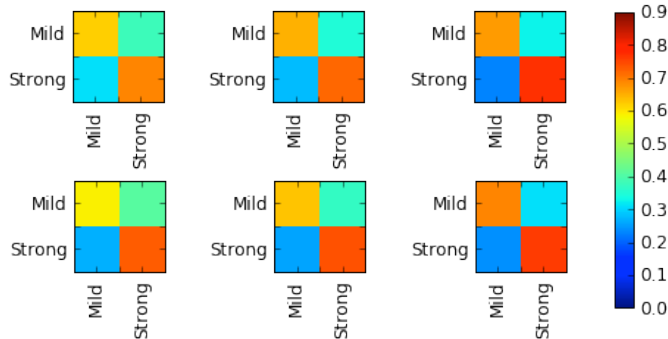

Figure 2: Estimated (top) and observed (bottom) confusion matrices for asthma dataset: These matrices correspond to the group of decision makers who have relatively little experience (# of years practising medicine = 0/1) and the group of patients with allergies but no past asthma attacks.

**Cluster Quality.** The prototype and the subspace feature indicator vector of a cluster allow us to understand the nature of the instances in the cluster. For instance, if the subspace feature indicator vector signifies that *gender* is the one and only important feature for some decision maker cluster $c$ and if the prototype of that cluster has value *gender = female*, then we can infer that $c$ constitutes of female decision makers. Since we are able to associate each cluster with such patterns, we can readily define the notions of *purity* and *inverse purity* of a cluster.

Consider cluster $c$ from the example above again. Since the defining pattern of this cluster is *gender = female*, we can compute the purity of the cluster by calculating what fraction of the decision makers in the cluster are *female*. Similarly, we can also compute what fraction of all decision makers who are *female* and are assigned to cluster $c$. This is referred to as inverse purity. While purity metric captures the notion of cluster homogeneity, the inverse purity metric ensures cluster completeness.

We compute the average purity and inverse purity metrics for the CoT, its ablation and a baseline JC across all the decision maker and item clusters and the results are presented in Table 3. Notice that CoT outperforms all the other ablations and the JC baseline. It is interesting to note that the non-interpretable CoT (NI-CoT) has much lower purity and inverse purity compared to non-temporal CoT (NT-CoT) as well as the CoT. This is partly due to the fact that the NI-CoT does not model prototypes or feature indicators which leads to less pure clusters.

**Qualitative Inspection of Insights.** We now inspect the cluster descriptions and the corresponding confusion matrices generated by our approach. Figure 2 shows one of the insights obtained by our model on the asthma dataset. The confusion matrices presented in Figure 2 correspond to the group of doctors with 0/1 years of experience evaluating patients who have allergies but did not suffer from previous asthma attacks. The three confusion matrices, one for each year (from left to right), shown on the top row in Figure 2 correspond to our estimates and those on the bottom row are computed from the data. It can be seen that the estimated confusion matrices match very closely with the ground truth thus demonstrating the effectiveness of the CoT framework.

Interpreting results in Figure 2, we find that doctors within the first year of their practice (left most confusion matrix) were recommending stronger treatments (nebulizers and immunomodulators) to patients who are likely to get better with milder treatments such as low impact drugs and inhalers. As time passed, they were able to better identify patients who could get better with milder options. This is a very interesting insight and we also found that such a pattern holds for client managers with relatively little experience. This could possibly mean that inexperienced decision makers are more likely to be risk averse, and therefore opt for safer choices.

## Footnotes

[1]Due to space limitations, we present the update equations assuming that the features of decision makers and items are discrete.

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
