[Supplementary Material · appendix.pdf]

# 1 Appendix

**Complete Generative Process of the CoT**   The generative process for the CoT can be summarized as follows: First, we generate the prototypes and the subspace feature indicator vectors for each of the decision maker and item clusters as follows:

$$p_c \sim Uniform(1, |J|) \; \forall c$$

$$p'_d \sim Uniform(1, |I|) \; \forall d$$

$$\omega_{c,m} \sim Bernoulli(\beta) \text{ where } \beta \sim Beta(\epsilon_\beta) \; \forall c, m$$

$$\omega'_{d,n} \sim Bernoulli(\beta') \text{ where } \beta' \sim Beta(\epsilon'_\beta) \; \forall d, n$$

We also sample the confusion matrices associated with every pair of (decision maker,item) clusters:

$$\Theta^{(t)}_{c,d,z}(e) \sim Dirichlet(\wedge) \; \forall c, d, z, \text{ if } t = 1$$

$$\Theta^{(t)}_{c,d,z}(e) \sim Dirichlet(h_{\Theta^{(t)}_{c,d,z}(e),\Gamma}) \text{ where } h_{\Theta^{(t)}_{c,d,z}(e),\Gamma} = \Gamma \left(1 + \pi \left[\Theta^{(t)}_{c,d,z}(e)\right]\right) \; \forall c, d, z, \text{ if } t \geq 2$$

where $z$ (row index) and $e$ (column index) jointly index an element of the confusion matrix.

We then sample the cluster assignments and features of decision makers and items.

$$c_j \sim Multinomial(\alpha) \text{ where } \alpha \sim Dirichlet(\epsilon_\alpha) \; \forall j$$

$$d_i \sim Multinomial(\alpha') \text{ where } \alpha' \sim Dirichlet(\epsilon'_\alpha) \; \forall i$$

Discrete features are sampled as:

$$a^{(j)}_m \sim Multinomial(\phi_{c_j,m}) \text{ where } \phi_{c_j,m} \sim Dirichlet(g_{p_{c_j,m},\omega_{c_j,m},\lambda}) \text{ and } g \text{ defined in Eqn. 1 } \; \forall j, m$$

$$b^{(i)}_n \sim Multinomial(\phi'_{d_i,n}) \text{ where } \phi'_{d_i,n} \sim Dirichlet(g_{p_{d_i,n},\omega_{d_i,n},\lambda}) \text{ and } g \text{ defined in Eqn. 1 } \; \forall i, n$$

In the case of continuous features, we have the following generative steps:

$$a^{(j)}_m \sim Normal(\phi_{c_j,m}, \sigma) \text{ where } \phi_{c_j,m} = p_{c,m} \text{ if } \omega_{c_j,m} = 1, \text{ Otherwise } \phi_{c_j,m} = 0 \; \forall j, m$$

$$b^{(i)}_n \sim Normal(\phi'_{d_i,n}, \sigma) \text{ where } \phi_{d_i,n} = p_{d,n} \text{ if } \omega_{d_i,n} = 1 \text{ Otherwise } \phi_{d_i,n} = 0 \; \forall i, n$$

The true labels associated with each of the items are sampled as:

$$z_i \sim Multinomial(\rho_{d_i}) \text{ where } \rho_{d_i} \sim Dirichlet(g'_{p'_{d_i}}) \text{ and } g' \text{ defined in Eqn. 2 } \; \forall i$$

Lastly, we sample the decisions made by decision makers on items along with the corresponding timestamps.

$$t_{j,i} \sim Uniform(1, T) \; \forall j, i$$

$$r_{j,i} \sim Multinomial(\Theta^{(t_{i,j})}_{c_j,d_i,z_i}) \; \forall j, i$$