[Reviews · NeurIPS 2016]

Reviewer 1

Summary

This paper provides a framework to model decision making trends from observed data. It extends existing frameworks in several ways, including modeling evolution over time and providing detailed results that help interpret and learn about the decision making process for specific groups of evaluators. At the core of the approach is a much more detailed model that groups evaluators and items into clusters, and models features of those clusters over time. The new framework is compared to all of the major existing framework and shown to outperform all of them on a consistent basis. It is also the only framework that can yield detailed insights about the decision making process, e.g. how the decisions of evaluators change over time.

Qualitative Assessment

This is an excellent paper in all aspects. 1) The writing style is excellent, which makes it easy to follow. 2) The idea of using a clustering approach, combined with inference of latent variables (e.g. true states) is novel and creative, and excellent fit for this type of application. 3) The resulting models may have societal impact, as they can help us better understand and learn about the the mistakes made in a wide variety of decision making. 4) The validation is excellent and comprehensive, comparing the proposed model to all major existing frameworks and using a wide variety of measures. All together, a pleasure to read. Kudos to the authors!

Confidence in this Review

2-Confident (read it all; understood it all reasonably well)


Reviewer 2

Summary

This paper describes a novel Bayesian model for the purpose of identifying confusions of evaluators with respect to evaluated item features, applications for which include any problem relating to individuals' decision-making based on a set of item features. The novel aspect of this approach is that it takes into account temporal aspects, i.e. the performance of individual evaluators over time. To achieve this individual evaluators and items are grouped into clusters, for which the algorithm then identifies a prototypical feature set based on which individual confusion matrices for evaluator/item combinations are determined across individual data points in time.

Qualitative Assessment

The proposed Confusions over Time algorithm (CoT) is promising, since it combines many aspects that have been treated in isolation by previous approaches (i.e. identifying error properties by evaluators, analysing expertise of evaluators, gaining insight into collective decision making). However, the central novelty is the inclusion of temporal aspects that reflect changes in decision quality. The algorithm is evaluated on three data sets and compared the performance to existing approaches with respect to the prediction of item labels, inference of confusion matrices and prediction of decisions. In addition, CoT ablations are tested to exclude the temporal aspects (NT-CoT) and the subspace feature generation within the individual clusters (NI-CoT). The results highlight the complementary benefit of the individual components, with particular improvement of the status quo for the inference of confusion matrices. The results are interpreted in an accessible manner. To allow comparison by future research, the used data sets should be explicitly identified and referenced (if publicly available). Generally, the paper provides a significant improvement over the status quo, is presented in an accessible manner and deserves visibility. I was missing a stronger final statement with respect to shortcomings of the current model and potential future developments. Some comments on presentation: - The Plate notation in Figure 1 is not visible when printed b/w. - The locations of Tables and Figures are sometimes confusing: Table 3 is only referenced on page 8, but appears on page 7. Figure 2 is on the correct page, but far from the textual reference. - Line 223: `... . Non-inte(r)pretable CoT ...' - Line 304: `... truth, thus demonstrating ... ' (comma missing) - Line 304: `effectively' should probably be `effectivity'? - Line 312: `... details, thus making ...' (comma missing)

Confidence in this Review

2-Confident (read it all; understood it all reasonably well)


Reviewer 3

Summary

This work builds a Bayesian model for classification using clusters and prototypes. It is motivated by building interpretable models and pieces together work from phenotyping, clustering and temporal modeling. While the formulation lacks substantial novelty, it does provide strong experimentation and results. More details about the data and stronger predictive baselines would strengthen this work.

Qualitative Assessment

The paper provides strong results on three data sets which goes alongside a thorough presentation of the model and the underlying motivations. The text could be more concise. More information about the data sets would be helpful, e.g. where they come from, feature extraction methods, class imbalance. My concerns experimentally are that the predictive baselines may be artificially weak, as logistic regression is likely to underperform given the number of features compared to the number of labels, and that the other models may not optimized for predictive tasks. A comparison to a semi-supervised learning method also seems appropriate. Furthermore, the time-slicing may artificially reduce the performance of the comparison methods. The synthetic data experimental in the third to last paragraph feels like an add-on and out of place. The conclusions in the final paragraph are perhaps overreaching given how little detail about the data sets was provided. --- Edit: After reading the reviews and author feedback, I have adjusted my score on novelty. The author feedback better illustrates the novelty of their work, however, I remain concerned about minimal discussion of many machine learning areas closely related to their work (e.g. see my original response and that of reviewers 4 and 6). I also have concerns about the exposition of their experiments (e.g. full experimental comparison to baselines).

Confidence in this Review

2-Confident (read it all; understood it all reasonably well)


Reviewer 4

Summary

An interpretable Bayesian model is developed which allows to characterize trends in decision making, in particular error patterns can be identified.

Qualitative Assessment

The model is pretty straightforward but seems to be more principled than previous approaches. It is simple and interpretable. It's usefulness is exemplified on some datasets.

Confidence in this Review

2-Confident (read it all; understood it all reasonably well)


Reviewer 5

Summary

This paper proposes a new Bayesian framework that diagnoses human decision making by jointly inferring error patterns of individual evaluators over time, true labels of items, and interpretable patterns underlying decision making process. Experimental results are shown for real world data on bail decisions, asthma treatments, and insurance policy approval decisions.

Qualitative Assessment

The paper mentions that a major drawback of the Dawid-Skene framework is that it does not provide any diagnostic insights into the collective decision making process. However, the authors do not provide enough convincing results to show that their model, CoT, can gain better and useful diagnostic insights. Also, the idea of representing clusters by prototypes and subspace feature indicators is not new and must have been explored by works on K-means. In general, the paper is well-written. However, more explanation on the model, especially for Figure 1, is needed.

Confidence in this Review

1-Less confident (might not have understood significant parts)


Reviewer 6

Summary

The authors propose a generative model for time-stamped K-categorical ratings specific to evaluator—item pairs. The authors cover relevant background work in the crowdsourcing and human annotation literature in section 2. In section 3, the authors describe the proposed CoT model which is based on clustering evaluators and items and associating a series of KxK confusion matrices with each (evaluator-cluster, item-cluster) pair. Each series contains T confusion matrices, one for each discrete time step in the data, and a simple Markov chain is imposed over the series so that the matrices evolve smoothly through time. In section 4, the authors describe a series of experiments where they compare the proposed model to ablations of the model and three existing baselines on prediction and estimation tasks using both real and synthetic data.

Qualitative Assessment

The authors motivate the proposed model with the setting in which items have “true” but unobserved labels/ratings and the observed labels/ratings given by evaluators are potentially incorrect. This differs from the very common problem in recommendation systems or collaborative filtering where evaluators provide their subjective ratings but there is not assumed to be any “true” rating (e.g., users of Netflix giving 1-5 star ratings to movies). This seems like a common but underexplored setting that is worthy of further study within machine learning. The authors are also right to highlight interpretability as a desired aspect of any machine learning solution that may yield post-hoc insights into common human biases and thus suggest corrective measures. This paper does a good job of motivating the proposed model and situating it within the crowdsourcing and human annotation literature. However the paper does not provide any coverage of two highly related areas of machine learning research. The proposed generative model is a stochastic block model (SBM) [1, 2]. There is a vast literature on SBMs that includes many dynamic variants. I think this paper must at least mention this literature to show that the proposed model is novel. Furthermore, the setting of evaluator—item categorical ratings is very familiar to machine learning within the areas of collaborative filtering, recommendation systems, and matrix factorization yet the paper does not mention any of these areas. Much of the writing is clear but the description of the model is not easy to read owing to some mistakes in the mathematical definitions and overly burdensome notation. The binomial distribution defined on line 128 and throughout should be the Bernoulli distribution. Similarly the multinomial distribution defined on line 138 and throughout should be the categorical distribution. The uniform distribution has support on the real numbers and thus the uniform distribution defined on line 124 and throughout should be a categorical distribution. Is V_{m,e} in equation 1 ever defined? The description of the experiments is vague. It is unclear where and when the true labels are used during training. On line 211, it’s stated that the true labels are only used for evaluation but on line 228 it’s said that 5% are used during training. It’s also not said how the Gibbs sampler is used to generate predictions. Is the last sample taken as a point estimate of the latent parameters? Or are multiple samples being used? It’s stated that the Gibbs sampler can be parallelized but not shown how. Also, how exactly is BIC used to set the number of clusters? With cross-validation? The experiments section in general needs to be more precise in order for the results to be understood and interpreted. [1] Nowicki, K. and Snijders, T. A. B. Estimation and prediction for stochastic blockstructures. Journal of the American Statistical Association, 96(455):1077–1087, 2001. [2] Kemp, C., Tenenbaum, J. B., Griffiths, T. L., Yamada, T., and Ueda, N. Learning systems of concepts with an infinite relational model. In Proceedings of the Twenty-First National Conference on Artificial Intelligence, 2006.

Confidence in this Review

2-Confident (read it all; understood it all reasonably well)